# Purification of Ethyl Linoleate from Foxtail Millet (*Setaria italica*) Bran Oil via Urea Complexation and Molecular Distillation

**DOI:** 10.3390/foods10081925

**Published:** 2021-08-19

**Authors:** Xiaoli Huang, Yuehan Zhao, Zhaohua Hou

**Affiliations:** College of Food Science and Engineering, Qilu University of Technology (Shandong Academy of Sciences), No. 3501 Daxue Road, Changqing District, Jinan 250353, China; huangxl99@163.com (X.H.); zhaoyuehan525@163.com (Y.Z.)

**Keywords:** foxtail millet (*Setaria italica*) bran oil, ethyl linoleate (ELA), urea complexation (UC), molecular distillation (MD), gas chromatography–mass spectrometry (GC–MS)

## Abstract

Foxtail millet (*Setaria italica*) bran oil is rich in linoleic acid, which accounts for more than 60% of its lipids. Ethyl linoleate (ELA) is a commercially valuable compound with many positive health effects. Here, we optimized two ELA processing steps, urea complexation (UC) and molecular distillation (MD), using single-factor and response surface analyses. We aimed to obtain a highly concentrated ELA at levels that are permitted by current regulations. We identified the optimal conditions as follows: 95% ethanol-to-urea ratio = 15:1 (*w*/*w*), urea-to-fatty acid ratio = 2.5:1 (*w*/*w*), crystallization time = 15 h, and crystallization temperature = −6 °C. Under these optimal UC conditions, ELA concentration reached 45.06%. The optimal MD purification conditions were established as follows: distillation temperature = 145 °C and vacuum pressure = 1.0–5.0 × 10^−2^ mbar. Under these conditions, ELA purity increased to 60.45%. Together, UC and MD were effective in improving the total concentration of ELA in the final product. This work shows the best conditions for separating and purifying ELA from foxtail millet bran oil by UC and MD.

## 1. Introduction

Foxtail millet (*Setaria italica*, abbreviated as FM) is one of the world’s oldest cultivated crops. It originated from China and, because of its excellent drought resistance, high tolerance to poor soil, and good nutrient value, it is now planted in India, Japan, Australia, North Africa, and South America [1]. FM possesses effective medical and nutritional qualities, including the ability to inhibit oxidants, cancer, and inflammation [2]. FM bran (FMB) is a by-product of FM processing and is produced after shelling [3]. It is rich in lipids, starch, proteins, vitamins, and minerals [4,5] and it is typically not fully utilized. Its current uses in China are mostly as animal feed and agricultural fertilizer. However, FMB should be viewed as a valuable resource that can be developed and utilized to produce value-added products. FMB utilization in this manner not only would effectively use an agricultural waste product but also might enrich farmers and improve their standard of living. For example, FMB may be a good source of linoleic acid, a potentially bioactive component.

Ethyl linoleate (ELA) is obtained by esterification of linoleic acid and ethanol, catalyzed by sulfuric acid. ELA has many physiological functions, such as enhancing immunity, reducing cholesterol and blood lipid levels [6], and mediating and controlling metabolism [7]. It is also the raw material for producing a drug that is highly effective at preventing and treating chronic diseases such as cerebral thrombosis and atherosclerosis [6]. It is also an effective treatment for mild to moderate acne, with effects on both inflammatory and non-inflammatory acne lesions [8]. In addition, ELA could also be added to health food [9], because of its function of reducing cholesterol and blood lipid levels. Hutsell, TC and Quackenbush, FW. suggest that subjects that received a fat-free diet with 2% of ELA showed lower plasma and hepatic cholesterol levels and more complete aortic protection than control subjects [10]. Therefore, ELA can be used as food for special medical purposes to help reduce plasma and hepatic cholesterol levels. Therefore, ELA has broad market applications in the pharmaceutical, cosmetic, health food, and other industries. It is thus a product that increases the value of FMB.

Several methods can be used to obtain polyunsaturated fatty acids (PUFA) from FMB; these include controlled low-temperature crystallization, winterization, urea complexation (UC), molecular distillation (MD), silver ion complexation, liquid chromatography, and extraction with supercritical carbon dioxide [11]. However, those methods are slow, inefficient, may generate undesirable by-products, and are generally difficult to scale up for industrial production [12].

The simplest and most efficient technique to obtain PUFAs is UC [13], a method that separates mixtures of fatty acids according to their degree of unsaturation [14]. Unsaturated fatty acids are non-linear molecules because of the presence of multiple double bonds, whereas saturated and monounsaturated fatty acids are almost linear, with a structure that can be easily trapped in the cavity of a urea complex [15]. The main advantage of UC is the extreme stability of the crystal complex, which obviates the need for filtration at the very low temperatures required for solvent crystallization of fatty acids [13].

MD is a gentle distillation method suitable for separating and purifying high-molecular-weight, thermally unstable materials and liquids at low vapor pressure and low temperature [16]. In the MD process, vapor molecules escape from the evaporator and are then captured in a condenser. Thus, it is necessary for vapor molecules to find a free path between the evaporator and the condenser. Material separation is achieved based on differences in the mean free path of a particle. Distillation occurs in a stationary state, where flows of two products are continuously generated; these consist of distillate cuts that are made of condensed molecules and the residue of MD, which is composed of non-evaporated molecules [17]. This technique reduces the risk of oxidative damage and is thus a very useful technique in the purification of thermolabile substances [15].

Many studies that used UC or MD to purify ELA focused on only one technique [13]. Here, we used both UC and MD to purify ELA from FMB oil. We determined the optimum ratios of 95% ethanol-to-urea and urea-to-fatty acid and determined the optimum crystallization time and temperature to separate ELA from FMB oil by UC. ELA was purified by MD. Gas chromatography–mass spectrometry (GC–MS) was used to compare fatty acid compositions before and after purification aand determine the optimal purification processes for ELA.

## 2. Materials and Methods

### 2.1. Materials

FM (*Setaria italica*) was purchased from Ji’nan, Shandong Province, China, on 15 June 2020. Only ripe (the ear seeds were completely hardened and presented the color of the original species), undamaged seeds were used. The millet was shelled, and the millet bran was obtained after dehulling (dehusking), milling, and polishing. Dehulling was done using a stone abrasive dehuller for 30 min, and bran was separated from the grain by winnowing.

A standard mixture of 21 fatty acid ethyl esters (FAEEs) was purchased from Nu-Chek-Prep. Inc. (Elysian, MN, USA). Hexane (chromatographic-grade) was purchased from the Beijing Reagent Company (Beijing, China). Aqueous ethanol, anhydrous Na_2_SO_4_ (analytical reagent, 99.0%), KOH (purity: >98.0%), and HCl (36%, AR) were acquired from Tianjin Guangfu Chemical Reagent Company (Tianjin, China). Petroleum ether (boiling range: 30–60 °C) was an analytical-grade, pure reagent purchased from the chemical reagent Co., Ltd. of the Chinese medicine group. Water was deionized using a Milli-Q purification system (Millipore, Schwalbach/Ts, Germany). All other solvents and chemicals used were of analytical or GC grade.

### 2.2. Sub-Critical Fluid Extraction (SFE) of Bran Oil and Preparation of Mixed Fatty Acid Ethyl Esters (MFAEs)

A method previously used to extract bran oil and prepare MFAEs was used in this study, with slight modifications [18]. Briefly, FMB oil was extracted using tetrane as a solvent. Single-factor and orthogonal experiments were used to optimize the extraction of FMB oil by SFE.

Fatty acid esters (FFAs) were obtained by esterifying FMB oil. A KOH solution was prepared by dissolving 56 g NaOH in 1 L of aqueous ethanol. A mixture containing 10 g of oil and 150 mL of 1N KOH–C_2_H_5_OH solution was heated at specified temperatures and times, and then the mixture was cooled to 20 °C. A solution of 15% NaCl was then added to the mixture. A solution of 20% hydrochloric acid was then added until the solution was at pH 2–3. Then, this mixture was put in a separatory funnel. Light petroleum was used for the extraction, for three times, and the extract was combined with the organic phase. The FFA-containing upper layer was dried with anhydrous sodium sulfate, and the solvent was evaporated in a vacuum rotary evaporator at 50 °C.

### 2.3. ELA Enrichment by UC

UC was performed following a previously described procedure with slight modifications [18]. Briefly, the bran oil ethyl ester was mixed with different amounts of urea-saturated 95% aqueous ethanol and fluxed until a homogenous solution was obtained. The resulting mixture was refluxed until its appearance became clear. The solution was then cooled to 0–4 °C and held at this temperature overnight to allow for crystallization to occur. The resulting crystals were collected by filtration, and the filtrate was concentrated in vacuo. The resulting residue was treated with 0.1 N HCl. The acidified solution was then extracted thrice by petroleum ether. The extract was dried over anhydrous sodium sulfate and distilled to dryness in vacuo. The resulting ELA concentrate was then stored at −4 °C.

#### 2.3.1. Single-Factor Test Design

Optimum 95% ethanol-to-urea ratio (*v*/*w*), urea-to-fatty acid ratio (*w*/*w*), crystallization temperature, and crystallization time based on the recovery and purity of ELA were determined using single-factor experiments. Variables and experimental levels for the single-factor tests were as follows: 95% ethanol-to-urea ratios (*v*/*w*) of 5:1, 10:1, 15:1, 20:1, and 25:1; urea-to-fatty acid ratios (*w*/*w*) of 1:1, 1:1.5, 1:2, 1:2.5, 1:3, 1:3.5, and 1:4; crystallization temperatures of −18, −15, −12, −9, −6, −3, 0, 3, and 6 °C; crystallization times of 3, 6, 9, 12, 15, 18, 21, 24, and 27 h. These parameters were established following a previous studies with slight modifications [17].

#### 2.3.2. Optimization of the Experimental Design

Single-factor experiments were used to evaluate the factors that influenced the results of ELA-containing liquid for further optimization. Purification conditions of ELA were optimized using a response surface Box–Behnken central composite design, using the purity of ELA as the response value. The coding of factors and levels of the response surface are shown in Table 1.

### 2.4. Wiped Film-Short Path MD Equipment and Methodology

The apparatus used in this study consisted of the KDL 5 short path MD system [19]. The schematic diagram for this equipment is shown in Figure 1. The apparatus was constructed mostly from glass. The evaporator was heated using a jacket through which heated oil from an oil bath was circulated. The vacuum system included a diffusion (10^−3 mbar^) and a mechanical (10^−2^ mbar) pump. The speed of the roller wiper inside the evaporator ranged from 50 to 400 rpm. An external condenser and a cold trap were installed immediately downstream of the still. Condensable, low-molecular-weight compounds were collected in the cold trap upstream of the vacuum system, and the cold trap was filled with liquid nitrogen to prevent volatile compounds from reaching the vacuum system. The operation of short-path MD is based on using proper temperature, pressure, and feed mass flow. The feed was placed inside a dosing vessel (8) after being heated and melted. Then, the vacuum pump was turned on to remove gases from within the feed. Once a steady vacuum pressure was achieved and the distillation temperature was fixed, the feeding valve was turned on, allowing the degassed feed liquid to immediately flow down. A scraper quickly spun it into a very thin film that spread over an evaporating surface. Due to the heated walls and high vacuum, the more volatile and light components were concentrated onto a closely positioned internal condensing surface, forming the distillate. The less volatile and heavy components flowed down along the cylinder, forming the residue. Thus, the resulting fractions were separated and discharged through their respective outlets. The yields were calculated after collecting and measuring the amounts of distillates and residues.

ELA was distilled under two different conditions using a short-path MD apparatus. Two-step distillations were performed by varying the evaporation temperature from 90 °C to 145 °C. The ELA-containing liquid was placed inside the dosing vessel (capacity, 500 mL), and the different fractions were separated by modifying the pressure and temperature while wiper speed and flow were kept constant. During the first distillation, the evaporating temperature, internal condenser temperature, and operating pressure were set at 90 °C, 15 °C, and 1.0–3.0 mbar, respectively, resulting in two fractions called Distillate 1 (DF_1_) and Residue 1 (RF_1_). In the second distillation, RF_1_ was distilled at evaporating temperature, internal condenser temperature, and operating pressure of 145 °C, 15 °C, and 0.01–0.05 mbar, respectively, resulting in two new fractions called Distillate 2 (DF_2_) and Residue 2 (RF_2_). The experimental conditions of the two processes are presented in Table 2.

### 2.5. Fatty Acid Analysis Using GC–MS

Fatty acids were analyzed by GC–MS using conditions described previously [20]. GC–electron impact ionization–MS analyses were performed using the 7000B GC–MS Triple Quad system (Agilent Technologies, Santa Clara, CA, USA) equipped with an SP^TM^-2560 fused silica capillary column (100 m × 0.25 mm × 0.25 μm; Supelco, PA, USA). The temperatures of the injector and transfer line were kept at 250 °C. The initial temperature of the oven was kept at 140 °C for 5 min and then increased by 4 °C/min to 200 °C; the temperature was increased again by 3 °C/min to 220 °C and held for 26 min. The injection volume was 1.0 μL, using helium as the carrier gas at a flow rate of 1.0 mL/min. The split mode was applied at a ratio of 60:1, and the solvent was delayed by 3 min. The electron energy was 70 eV, and the temperature of the ion source was set to 230 °C. Results were recorded in GC–MS full-scan mode at a resolution of *m/z* 50–500. Two groups (Foxtail millet bran oil and RF_2_) were analyzed and then divided into three replicates for every group. ELA purity was analyzed by GC–MS according to the above conditions and calculated by the peak area normalization method.

## 3. Results

### 3.1. Single-Factor Experiment on UC

#### 3.1.1. Optimization of 95% Ethanol/Urea Ratio

The effects of 95% ethanol/urea ratios of 5:1, 10:1, 15:1, 20:1, and 25:1 on oil purity were studied (Figure 2a). Increasing the 95% ethanol/urea ratio from 5:1 to 15:1 resulted in increased purity. At a 95% ethanol/urea ratio of 15:1, the purity of bran oil peaked at 37.99%, while at a 95% ethanol/urea ratio of over 15:1, the purity began to decrease. Therefore, this single-factor test suggested that the best ethanol concentration was 15:1.

#### 3.1.2. Optimization of the Urea/FAEE Ratio

The effects of urea/FAEE ratios of 1:1, 1.5:1, 2:1, 2.5:1, 3:1, 3.5:1, and 4:1 on oil purity are shown in Figure 2b. The crystallization time and temperature were 15 h and −6 °C, respectively. The maximum purity (40.22%) of ELA was obtained at the urea/FAEE ratio (*w*/*w*) of 4:1. However, the level of purity increased slowly starting from the ratio of 2.5:1. Therefore, the ratio of 4:1 is not suitable for large-scale production, and the urea/FAEE ratio of 2.5:1 is optimal based on our single-factor test.

#### 3.1.3. Effect of Improving Crystallization Time on the Purity

Figure 2c shows the effect of crystallization time on the purity of ELA at a urea/FAEE ratio 2.5:1 and a crystallization temperature of −6 °C. Purity increased slowly from 3 to 27 h. However, at crystallization times ranging from 15 to 27 h, ELA purity did not increase significantly. Thus, most ELA was extracted in 15 h. Hence, the optimal crystallization time appeared to be 15 h based on our single-factor test.

#### 3.1.4. Effect of Crystallization Temperature

The effect of crystallization temperature on ELA purity at a urea/FAEE ratio of 2.5:1 and a crystallization time of 15 h is shown Figure 2d. Because the process includes exothermal activities, low temperatures facilitate the crystallization of highly enriched PUFA. The purity of ELA increased as the temperature increased from −18 to −12 °C. However, from −12 to 6 °C, the purity decreased, except for a minor increase at −6 °C. The purity of ELA decreased quickly from 38.33% to 32.28% as the temperature increased from −6 to 6 °C, while it remained over 38% without significant changes at temperatures between −12 and −6 °C. Therefore, the optimum extraction temperature appeared to be −6 °C, based on our single-factor test results.

### 3.2. Optimization of UC Process Parameters by the Response Surface Method

#### 3.2.1. Response Surface Test Results

According to the design specifications of the center combination test of Box–Behnken in Design-Expert 7.0, the coefficients of the following independent variables were established as follows: 95% ethanol/urea ratio [X1]; urea/FAEE ratio [X2]; crystallization time [X3]; crystallization temperature [X4]. The design program and results of the response surface are shown in Table 3.

#### 3.2.2. Regression Analysis and Analysis of Variance

A central composite factorial design consisting of 29 experiments was implemented to examine the combined effects of medium components on ELA purity. The *p*-values were used as the tool to check the significance of each variable. This statistical analysis was used to understand the pattern of mutual interactions between the selected variables. The regression equation was evaluated using an F-test. Analysis of variance (ANOVA) results (Table 4) of the optimization study indicated that X_1_, X_2_, X_2_X_3_, X_2_X_4_, X_1_^2^, X_2_^2^, X_3_^2^, and X_4_^2^ were significant model terms (*p* < 0.05). Among the variables tested in the present study, the effects of 95% ethanol/urea ratio (X1) and urea/FAEE ratio (X2) were independently significant. The interaction effects between urea/FAEE ratio (X2) and crystallization time (X3) and between urea/FAEE ratio (X2) and crystallization temperature (X4) were also significant. The F-value for lack of fit was non-significant, indicating that the model was a good fit [21,22].

The lack of fit was not significant, and R^2^ was 0.9749, which indicated that the regression equations were good and the linear relationship between the independent variable and the response value was significant; thus the model could explain 97.49% of the variation in response. The R_adj_^2^ value was 0.9498, indicating that the model was acceptable. The R^2^ value of a good statistical model should be in the range of 0–1.0, with R^2^ values near 1.0 indicate better degrees of fit [23].

The regression equation coefficients were calculated, and the data were fitted to a second-order polynomial equation. The regression equation for purity dependent on the 95% ethanol/urea ratio [X1], urea/FAEE ratio [X2], crystallization time [X3], and crystallization temperature [X4] is as follows:Y_1_ = + 41.73 − 1.33X_1_ − 2.18X_2_ + 0.28X_3_ − 0.36X_4_ + 0.91X_1_X_2_ + 0.13X_1_X_3_ + 1.01X_1_X_4_ − 1.63X_2_X_3_ − 2.33X_2_X_4_ − 0.66X_3_X_4_ − 6.23X_1_^2^ − 11.39X_2_^2^ − 4.40X_3_^2^ − 6.12X_4_^2^(1)

Two of the four factors were fixed at the zero level, and the other two factors and ELA purity (Y) were used as the three-dimensional response surface diagrams and contour map (Figure 3). The optimal process parameters were identified from the central point of the contour plot.

Mathematical analysis based on the response surface showed that the optimum purification conditions were as follows: the 95% ethanol/urea ratio was 2.327:1, the urea/FAEE ratio was 15.937:1, the crystallization time was 15.606 h, and the crystallization temperature was −6.267 °C. The maximum purity of ELA predicted by the regression model was 41.934%.

#### 3.2.3. Validation of the Predictive Model

Three parallel verification experiments were conducted to validate the best experimental conditions as predicted by the above model. According to the actual situation, the optimum conditions were as follows: 95% ethanol/urea ratio of 2.5:1, urea/FAEE ratio of 15:1, crystallization time of 15 h, and crystallization temperature of −6 °C. The model predicted an ELA purity of 41.934%. To verify the suitability of the model for predicting optimum response values, we performed an optimization experiment, which produced ELA with a purity of 42.241% under optimal conditions. The real and predicted values did not differ significantly (*p* < 0.05), indicating that the model was adequate for the process.

### 3.3. Molecular Distillation

The ELA mixture was purified by MD using conditions (Table 1) optimized during a preliminary experiment. The distillation yields are present in Table 5. One residue (RF_2_) and two distillate (DF_1_ and DF_2_) fractions were prepared by MD. The yields of the distilled (DF_1_ and DF_2_) and residue (RF_2_) fractions were 8.22%, 56.87%, and 31.92% (*w*/*w*), respectively. The oil was placed inside a dosing vessel (capacity, 500 mL), and the different fractions were obtained by modifying the pressure and temperature, while maintaining constant wiper speed and flow. During the first distillation, the evaporating temperature, internal condenser temperature, and operating pressure were set to 90 °C, 15 °C, and 1.0–3.0 mbar, respectively, and the oil feed weighed 376.33 g (400 g). After the first distillation, the resulting distillate (DF_1_) and residue (RF_1_) fractions were obtained with yields of 8.22% and 89.89%, respectively. Part of RF_1_ was used to feed the second distillation process at 145 °C, 15 °C, and 0.01–0.05 mbar, resulting in new distillate (DF_2_) and residue (RF_2_) fractions.

### 3.4. MFAE Composition of Bran Oil and RF_2_

The MFAE compositions of the bran oil and RF_2_ are shown in Table 6. The area and concentration values listed in Table 6 are averages of three measurements. The following eight fatty acids were identified in both bran oil and RF_2_: palmitic acid (C16:0), hexadecenoic acid (C16:1), stearic acid (C18:0), oleic acid (C18:1), linoleic acid (C18:2), linolenic acid (C18:3), arachidic acid (C20:0), and behenic acid (C22:0). Concentrations and compositions of the chemical components and fatty acids in the different fractions differed significantly.

Linoleic (18:2), stearic (18:1), and palmitic (16:0) acids accounted for 39.98%, 21.65%, and 10.01%, respectively, of the fatty acid ethyl esters of bran oil. These results are consistent with those of Shi Y.Z. et al. [1]. Linoleic, oleic, and linolenic acids were found to be the main unsaturated fatty acids in FMB, while palmitic and stearic acids appeared to be the main saturated fatty acids. Polyunsaturated fatty acids such as linoleic (18:2) and linolenic (18:3) acid are considered essential fatty acids because they are required by the human body [24] and are thus very important for human health. Our results suggest that FMB oil is a good source of pharmaceutical ELA.

Table 6 shows that bran oil and RF_2_ contained 41.98% and 60.45% ELA, respectively. The ELA content increased by 18.47% after UC and MD. Meanwhile, bran oil and RF_2_ contained 26.83% and 9.88% of saturated fatty acids (SFA), respectively. The SFA content decreased by 2.72-fold after UC and MD, indicating that UC and MD can significantly decrease the SFA content of bran oil.

## 4. Conclusions

In this study, a new method for separating and purifying ELA from FMB was optimized. The method uses UC and MD to obtain highly concentrated ELA. The optimal conditions, in UC process, identified via single-factor and response surface experiments were established as follows: 95% ethanol/urea ratio = 15:1 (*w*/*w*), urea/fatty acid ratio = 2.5:1 (*w*/*w*,; crystallization time = 15 h, and crystallization temperature = −6 °C. Under the optimal conditions, ELA concentration was 42.241%. We showed that response surface experiments can be used for regression analysis and parameter optimization of the UC process of foxtail millet bran oil. The ELA mixture was purified by MD in either one or two stages, and the best results were obtained using a one-stage purification with a distillation temperature of 145 °C and a pressure of 1.0–5.0 × 10^−2^ mbar. Under these conditions, ELA was purified to a level of 60.45%. By using both UC and MD, we were able to improve the total concentration of ELA in the final product. Therefore, ELA can be purified by the above method. The purification of ELA from FMB provides a resource for the field of health food and food for special medical purpose and adds value to subsidiary agricultural products.

## Figures and Tables

**Figure 1 foods-10-01925-f001:**
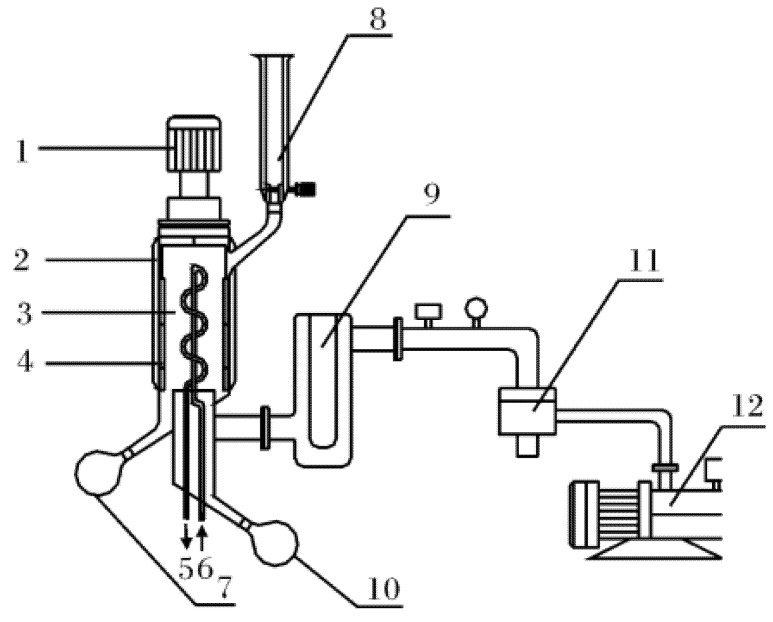
Apparatus diagram of the VKL 70 molecular system. 1, Motor drive; 2, heating jacket; 3, condenser; 4, wiper; 5, cooling water outlet; 6, cooling water inlet; 7, residue receiver; 8, dosing vessel; 9, cold trap; 10, distillate receiver; 11, diffusion pump; 12, rotary vane vacuum pump.

**Figure 2 foods-10-01925-f002:**
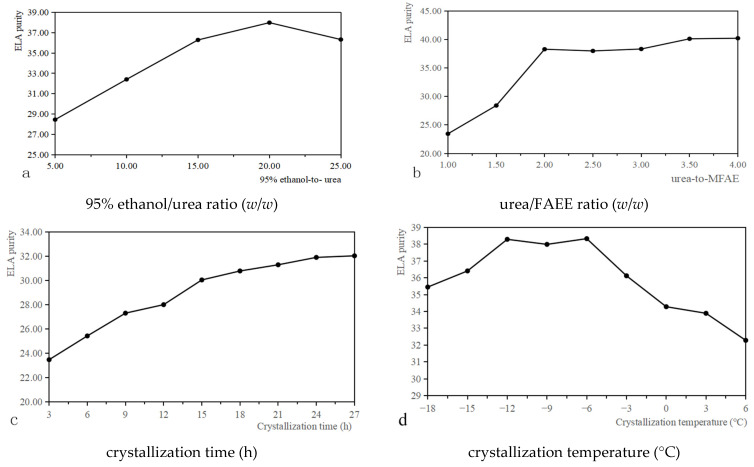
The effect of 95% ethanol/urea ratio (**a**) and urea/FAEE ratio (**b**), crystallization time (**c**), crystallization temperature (**d**) on ELA purity.

**Figure 3 foods-10-01925-f003:**
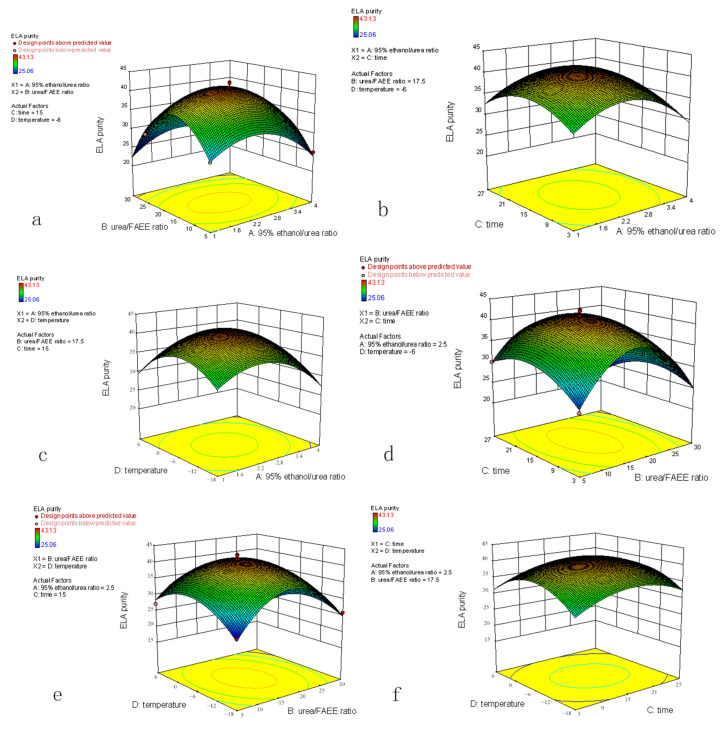
Response surface of ELA purity showing the mutual interactions of independent parameters. (**a**) The 95% ethanol/urea ratio (X1, *w*/*w*) and urea/FAEE ratio temperature (X2, *w*/*w*); (**b**) the 95% ethanol/urea ratio (X1, *w*/*w*) and crystallization time (X3); (**c**) the 95% ethanol/urea ratio (X1, *w*/*w*) and crystallization temperature (X4); (**d**) the urea/FAEE ratio temperature (X2, *w*/*w*) and crystallization time (X3); (**e**) the urea/FAEE ratio temperature (X2, *w*/*w*) and crystallization temperature (X4); (**f**) the crystallization time (X3) and crystallization temperature (X4).

**Table 1 foods-10-01925-t001:** Coding of factors and levels of response surfaces.

Independent Variables	Code Variable Levels		
	−1	0	1
95% ethanol-to-urea ratio (*v*/*w*) (g/g) (X1)	5	15	25
urea-to-fatty acid ratio (*w*/*w*) (g/g) (X2)	1	2.5	4
Crystallization time (h) (X3)	3	15	27
Crystallization temperature (°C) (X4)	−18	−6	6

**Table 2 foods-10-01925-t002:** Experimental conditions for the molecular distillation processes.

Experimental Condition		Value
Preheating temperature (°C)		35 ± 0.5
Wiper speed (rpm)		220–240
Feed rate (mL/min)		10.0 ± 1.0
The first level distillation process		
	Evaporating temperature (°C)	90 ± 0.2
	Internal condenser temperature (°C)	15 ± 1.0
	Operating pressure (mbar)	1.0–3.0
The second level distillation process		
	Evaporating temperature (°C)	145 ± 0.2
	Internal condenser temperature (°C)	15 ± 1.0
	Operating pressure (mbar)	0.01–0.05

**Table 3 foods-10-01925-t003:** Design program and results of response surface for UC.

Run	X1	X2	X3	X4	ELA Purity (%)
1	0	1	1	0	31.08
2	−1	−1	0	0	28.48
3	−1	0	0	−1	32.39
4	−1	1	0	0	29.84
5	0	0	1	−1	31.92
6	1	−1	0	0	25.09
7	0	−1	0	−1	25.06
8	0	−1	1	0	30.01
9	0	0	−1	1	32.13
10	0	1	−1	0	31.77
11	1	0	−1	0	29.95
12	0	0	0	0	40.49
13	0	0	−1	−1	29.01
14	0	1	0	1	28.28
15	1	1	0	0	29.37
16	0	0	0	0	41.02
17	−1	0	0	1	30.58
18	−1	0	−1	0	32.61
19	1	0	0	−1	25.6
20	1	0	0	1	27.84
21	0	−1	0	1	27.1
22	0	0	0	0	42.93
23	0	−1	−1	0	25.47
24	0	0	0	0	43.13
25	0	0	1	1	32.4
26	−1	0	1	0	32.48
27	1	0	1	0	30.33
28	0	0	0	0	41.72
29	0	1	0	−1	25.88

**Table 4 foods-10-01925-t004:** Regression coefficients of the predicted quadratic polynomial model for response variables (ELA purity) in urea inclusion fractionation experiment.

Source	Sum of Squares	df	Mean Square	F Value	*p*-Value	Notability
Model	784.10	14	56.01	38.81	<0.0001	***
X1	17.99	1	17.99	12.47	0.0033	***
X2	31.92	1	31.92	22.12	0.0003	***
X3	0.79	1	0.79	0.55	0.4713	
X4	1.36	1	1.36	0.94	0.3477	
X1X2	2.13	1	2.13	1.48	0.2443	
X1X3	0.065	1	0.065	0.045	0.8349	
X1X4	4.10	1	4.10	2.84	0.1140	
X2X3	6.84	1	6.84	4.74	0.0471	*
X2X4	16.29	1	16.29	11.29	0.0047	***
X3X4	1.74	1	1.74	1.21	0.2904	
X1X1	248.17	1	248.17	171.99	<0.0001	***
X2X2	430.83	1	430.83	298.57	<0.0001	***
X3X3	124.07	1	124.07	85.98	<0.0001	***
X4X4	248.68	1	248.68	172.34	<0.0001	***
Residual	20.20	14	1.44			
Lack of Fit	14.84	10	1.48	1.11	0.5020	
Pure Error	5.36	4	1.34			
Cor Total	804.30	28				

***: significant (*p* < 0.001); *: (*p* < 0.05). R^2^ = 0.9749; R_adj_^2^ = 0.9498.

**Table 5 foods-10-01925-t005:** Yields from multiple molecular distillation processes.

Distillation Process		Feed	Residue Fraction (RF)	Distillate Fraction (DF)
First process				
	Weight (g)	376.33 ± 2.62	338.04 ± 0.91 (RF_1_)	34.82 ± 2.45 (DF_1_)
	Yield (%, *w*/*w*)		89.89 ± 0.57%	8.22 ± 0.17%
Second process				
	Weight (g)	338.04 ± 0.91 (RF_1_)	114.34 ± 1.04 (RF_2_)	214.30 ± 1.88 (DF_2_)
	Yield (%, *w*/*w*)		31.92 ± 0.44%	56.87 ± 0.49%
Total weight (g)		363.66 ± 1.88		
Recovery rate (%, *w*/*w*)		96.63 ± 1.03%		

**Table 6 foods-10-01925-t006:** Contents of the components of foxtail millet bran oil and RF_2_.

Nr.	Fatty Acid	Molecular Formula	Foxtail Millet Bran Oil	RF_2_
			Ratio/%	Ratio/%
1	Palmitic acid ethyl ester (C16:0)	C_18_H_36_O_2_	10.01 ± 0.26	3.42 ± 0.39
2	Hexadecenoic acid ethyl ester (C16:1)	C_18_H_34_O_2_	6.22 ± 0.03	6.01 ± 1.04
3	Stearic acid ethyl ester (C18:0)	C_20_H_40_O_2_	9.28 ± 1.17	2.14 ± 0.44
4	Oleic acid ethyl ester (C18:1)	C_20_H_38_O_2_	21.65 ± 1.10	18.92 ± 0.19
5	Linoleic acid ethyl ester (C18:2)	C_20_H_36_O_2_	39.98 ± 1.83	60.45 ± 0.43
6	Linolenic acid ethyl ester (C18:3)	C_20_H_34_O_2_	5.24 ± 0.33	3.46 ± 0.41
7	arachidic acid ethyl ester (C20:0)	C_22_H_44_O_2_	5.03 ± 0.64	2.98 ± 0.86
8	Behenic acid methyl ester (C22:0)	C_24_H_48_O_2_	2.51 ± 0.09	1.34 ± 0.07

## Data Availability

The datasets generated for this study are available on request to the corresponding author.

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
