# Peer review of "Purification of Ethyl Linoleate from Foxtail Millet (Setaria italica) Bran Oil via Urea Complexation and Molecular Distillation"

_foods, 2021, doi:10.3390/foods10081925_

Round 1

Reviewer 1 Report

The manuscript is dealing with interesting topic which, as stated by authors, already exist in the literature. I believe however, that since it presents both methods it proves its scientific soundness.

Below I put some remarks that can help to improve the quality of paper:

line 82: how the maturation level of seeds was assessed?

line 83: how these steps (dehulling, milling, polishing) were performed?

line 116: italic for in vacuo?

line 114: how filtration was perforemd?

line 132:  why this range of parameters has been studied?

line 183: Please provide the methodology for statistical analysis and the number of replicates. 

Reviewer 2 Report

The manuscript describe a new method for separating and purifying ELA from FMB. It is very well conducted and written.

My only suggestion to reference: please correct the numeration of refernces ( for example: "1. 1. Shi Y.-Z.; Ma Y...") and the references 11 and 16 - there are the same.  

Reviewer 3 Report

The manusript entitled: “Purification of Ethyl Linoleate in Foxtail Millet (Setaria italica) Bran Oil via Urea Complexation and Molecular Distillation” reports information and data on ethyl linleate. ]

The main drawback is that the molecule obiect of the proposed study is mainly used in cosmetics and not in food area. Authors please define the end points and application in the food area asssessing use and safety aspects in the Introduction section.

At line 38 it is tated that: “Linoleic acid is commercially available in the form of ethyl linoleate”: please substantiate this sentence since linoleic acid interest i salso increasingly popular in the beauty products industry. Linoleic acid (18:2ω6; cis, cis-9,12-octadecadienoic acid) is the among themost highly consumed poly unsaturated fatty acids commonly used in the human diet. Among the main dietary sources of linoleic acid, which consider among the essential nutrients for itsbeneficial health roperties, are vegetable oils, nuts, seeds, meats, and eggs.

Lines 46-48 should be substatntiated by literature reference and data. The manuscript is interesting, nonetheless it is not clear if the scope is to obtain the linoleic acid or the ethyl linoleate. The end points of the manuscript should be better cleared (see Abstract and following). The ethyl linoleate use and application in the food area should be assessed and described. The experimental procedures used should better detailed and the overall experimental design better described. Limits and possible applicatin of the proposed study should be put in better evidence.

The overall writing seems confused: please check carefully the English language form and phrasing.

Round 2

Reviewer 3 Report

The manuscript has been modified and improved, nonetheless there is still poor clarity regarding the connection between ethyl linoleate and linoleic acid. The food application should be better addressed as well as the interest of ethyl linoleate in the food aspects. It has activity, nonetheless please exploit better its use and justify. The Conclusion section stilll stresses the results obtained on the ethyl linoleate wich is the main end point but for the Journal Foods, the limits and possible application in the foodstuff area of interest should be better described. Please add comments in this respect and add comment and Autors point of view in the Conclusion section. For example, please justify line 38 where it is stated that linoleic acid is "mostly available in the form of ethyl linoleate". Linoleic acid can be recovered from many food matrices. The food related aspects and applications should be better detailed regarding the ethyl linoleate.
